# ENet: A Deep Neural Network Architecture for Real-Time Semantic Segmentation

**Adam Paszke**
Faculty of Mathematics, Informatics and Mechanics
University of Warsaw, Poland
`a.paszke@students.mimuw.edu.pl`

**Abhishek Chaurasia, Sangpil Kim & Eugenio Culurciello**
Electrical and Computer Engineering
Purdue University, USA
`aabhish, sangpilkim, euge@purdue.edu`

## Abstract

The ability to perform pixel-wise semantic segmentation in real-time is of paramount importance in practical mobile applications. Recent deep neural networks aimed at this task have the disadvantage of requiring a large number of floating point operations and have long run-times that hinder their usability. In this paper, we propose a novel deep neural network architecture named ENet (efficient neural network), created specifically for tasks requiring low latency operation. ENet is up to $18\times$ faster, requires $75\times$ less FLOPs, has $79\times$ less parameters, and provides similar or better accuracy to existing models. We have tested it on CamVid, Cityscapes and SUN datasets and report on comparisons with existing state-of-the-art methods, and the trade-offs between accuracy and processing time of a network. We present performance measurements of the proposed architecture on embedded systems and suggest possible software improvements that could make ENet even faster.

## 1 Introduction

Recent interest in augmented reality wearables, home-automation devices, and self-driving vehicles has created a strong need for semantic-segmentation (or visual scene-understanding) algorithms that can operate in real-time on low-power mobile devices. These algorithms label each and every pixel in the image with one of the object classes. In recent years, the availability of larger datasets and computationally-powerful machines have helped deep convolutional neural networks (CNNs) (LeCun & Bengio (1998); Krizhevsky et al. (2012); Simonyan & Zisserman (2014a); Szegedy et al. (2015a)) surpass the performance of many conventional computer vision algorithms (Shotton et al. (2009); Perronnin et al. (2010); van de Sande et al. (2011)). Even though CNNs are increasingly successful at classification and categorization tasks, they provide coarse spatial results when applied to pixel-wise labeling of large images. Therefore, they are often cascaded with other algorithms to refine the results, such as color based segmentation (Farabet et al. (2013)) or conditional random fields (Chen et al. (2014)), to name a few.

In order to both spatially classify and finely segment images, several neural network architectures have been proposed, such as SegNet (Badrinarayanan et al. (2015a;b)) or fully convolutional networks (Long et al. (2015)). All these works are based on a VGG16 (Simonyan & Zisserman (2014b)) architecture, which is a very large model designed for multi-class classification. These references use models with a large number of parameters, and slow inference time. In these conditions, they become unusable for many mobile or battery-powered applications, which require processing images at rates higher than 10 fps.

In this paper, we propose a new neural network architecture optimized for high-accuracy and also fast inference. In our work, beside neural network processing, we chose not to use any other post-processing steps, in order to focus on the intrinsic performance of an end-to-end CNN approach.

In Section 3 we propose a fast and compact encoder-decoder architecture named ENet. It has been designed according to rules and ideas that have appeared in the literature recently, all of which we discuss in Section 4. Performance of the proposed network has been tested on Cityscapes (Cordts et al. (2016)) and CamVid (Brostow et al. (2008)) for driving scenario, whereas SUN dataset (Song et al. (2015)) has been used for testing our network in an indoor situation. We benchmark it on NVIDIA Jetson TX1 Embedded Systems Module as well as on an NVIDIA Titan X GPU. The results can be found in Section 5.

## 2 RELATED WORK

Semantic segmentation is important in fully understanding the content of images, find target objects and segment them. This technique is of utmost importance in applications such as driving and augmented reality. Moreover, real-time operation is a must for these applications, and therefore, designing CNNs *carefully* is vital. Contemporary computer vision applications extensively use deep neural networks, now one of the most widely used techniques for many different tasks, including semantic segmentation. This work presents a fully trainable neural network architecture, and therefore we aim to compare to other literature that performs the large majority of inference in the same way.

State-of-the-art scen-parsing CNNs use two separate neural network architectures combined together: an encoder and a decoder. Inspired by probabilistic auto-encoders (Ranzato et al. (2007); Ngiam et al. (2011)), encoder-decoder network architecture have been introduced in SegNet-basic (Badrinarayanan et al. (2015a)), and further improved in SegNet (Badrinarayanan et al. (2015b)). The encoder is a vanilla CNN (such as VGG16 from Simonyan & Zisserman (2014b)) which is trained to classify the input, while the decoder is used to upsample the output of the encoder (Long et al. (2015); Noh et al. (2015); Zheng et al. (2015); Eigen & Fergus (2015); Hong et al. (2015)). However, these networks are slow during inference due to their large architectures and numerous parameters. Unlike in Noh et al. (2015), fully connected layers of VGG16 were discarded in the latest incarnation of SegNet, in order to reduce the number of operations and memory footprint, making it the smallest of these networks. Still, none of them can operate in real-time.

Other existing architectures use simpler classifiers and then cascade it with Conditional Random Field (CRF) as a post-processing step (Chen et al. (2014); Sturgess et al. (2009)). As explained in Badrinarayanan et al. (2015b), these techniques use onerous post-processing steps and often fail to label the classes that occupy fewer number of pixels in a frame. CNNs can be combined with recurrent neural networks (Zheng et al. (2015)) for better performance, but suffers from further speed degradation. Also, one has to keep in mind that RNN, used as a post-processing step, can be used in conjunction with any other technique, including the one presented in this work.

## 3 NETWORK ARCHITECTURE

The architecture of our network is presented in Table 1. It is divided into several stages, as highlighted by horizontal lines in the table and the first digit after each block name. Output sizes are reported for an example input image resolution of $512 \times 512$. We adopt a view of ResNets (He et al. (2015b)) that describes them as having a single main branch and extensions with convolutional filters that separate from it, and then merge back with an element-wise addition, as shown in Figure 1b. Just as in the original paper, we refer to these as bottleneck modules. They consist of three convolutional layers: a $1 \times 1$ projection that reduces the dimensionality, a main convolutional layer (`conv` in Figure 1b), and a $1 \times 1$ expansion. If the bottleneck is downsampling, a max pooling layer is added to the main branch. We zero pad the activations, to match the number of feature maps. Also, the first $1 \times 1$ projection is replaced with a $2 \times 2$ convolution with stride 2 in both dimensions. `conv` is either a regular, dilated or full convolution (also known as deconvolution or fractionally strided convolution) with $3 \times 3$ filters. Sometimes we replace it with asymmetric convolution i.e. a sequence of $5 \times 1$ and $1 \times 5$ convolutions. For the regularizer, we use Spatial Dropout (Tompson et al. (2015)), with $p = 0.01$ before bottleneck2.0, and $p = 0.1$ afterwards.

The initial stage contains a single block, that is presented in Figure 1a. Stage 1 consists of 5 bottleneck blocks, while stage 2 and 3 have the same structure, with the exception that stage 3 does not downsample the input at the beginning (we omit the 0th bottleneck). These three first stages are the encoder. Stage 4 and 5 belong to the decoder.

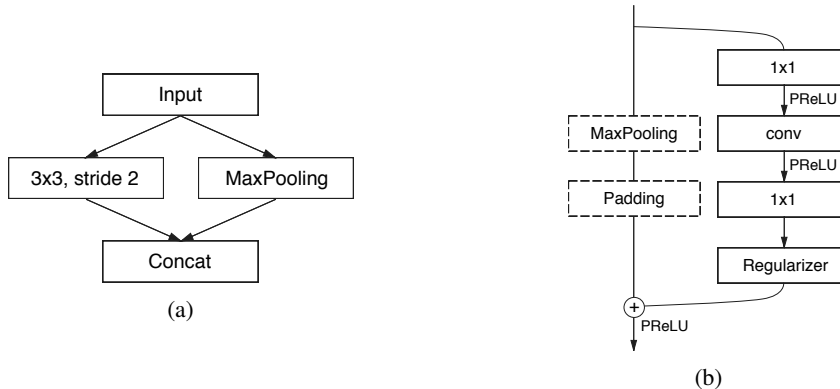

Figure 1: (a) ENet initial block. MaxPooling is performed with non-overlapping $2 \times 2$ windows, and the convolution has 13 filters, which sums up to 16 feature maps after concatenation. This is heavily inspired by Szegedy et al. (2015b). (b) ENet bottleneck module. `conv` is either a regular, dilated, or full convolution (also known as deconvolution) with $3 \times 3$ filters, or a $5 \times 5$ convolution decomposed into two asymmetric ones.

We did not use bias terms in any of the projections, in order to reduce the number of kernel calls and overall memory operations, as cuDNN (Chetlur et al. (2014)) uses separate kernels for convolutions and bias addition. This choice didn't have any impact on the accuracy. Between each convolutional layer and following non-linearity we use Batch Normalization (Ioffe & Szegedy (2015)). In the decoder max pooling is replaced with max unpooling, and padding is replaced with spatial convolution without bias. We did not use unpooling information in the last upsampling module, because the initial block operated on the 3 channels of the input frame, while the final output has $C$ feature maps (the number of object classes). Also, for performance reasons, we decided to place only a bare full convolution as last module of the network, which alone takes up a sizeable portion of the decoder processing time.

Table 1: ENet architecture. Output sizes are given for an example input of $512 \times 512$.

| Name | Type | Output size |
|---|---|---|
| initial | | $16 \times 256 \times 256$ |
| bottleneck1.0 | downsampling | $64 \times 128 \times 128$ |
| $4\times$ bottleneck1.x | | $64 \times 128 \times 128$ |
| bottleneck2.0 | downsampling | $128 \times 64 \times 64$ |
| bottleneck2.1 | | $128 \times 64 \times 64$ |
| bottleneck2.2 | dilated 2 | $128 \times 64 \times 64$ |
| bottleneck2.3 | asymmetric 5 | $128 \times 64 \times 64$ |
| bottleneck2.4 | dilated 4 | $128 \times 64 \times 64$ |
| bottleneck2.5 | | $128 \times 64 \times 64$ |
| bottleneck2.6 | dilated 8 | $128 \times 64 \times 64$ |
| bottleneck2.7 | asymmetric 5 | $128 \times 64 \times 64$ |
| bottleneck2.8 | dilated 16 | $128 \times 64 \times 64$ |
| *Repeat section 2, without bottleneck2.0* | | |
| bottleneck4.0 | upsampling | $64 \times 128 \times 128$ |
| bottleneck4.1 | | $64 \times 128 \times 128$ |
| bottleneck4.2 | | $64 \times 128 \times 128$ |
| bottleneck5.0 | upsampling | $16 \times 256 \times 256$ |
| bottleneck5.1 | | $16 \times 256 \times 256$ |
| fullconv | | $C \times 512 \times 512$ |

## 4 DESIGN CHOICES

In this section we will discuss our most important experimental results and intuitions, that have shaped the final architecture of ENet.

**Feature map resolution:** Downsampling images during semantic segmentation has two main drawbacks. Firstly, reducing feature map resolution implies loss of spatial information like exact edge shape. Secondly, full pixel segmentation requires that the output has the same resolution as the input. This implies that strong downsampling will require equally strong upsampling, which increase model size and computational cost. The first issue has been addressed in Long et al. (2015) by adding the feature maps produced by encoder, and in SegNet (Badrinarayanan et al. (2015a)) by saving the elements index from the corresponding encoder max-pooling module. We followed the SegNet approach, because it allows to reduce memory requirements, but we found that using a strong downsampling still reduces the final accuracy.

However, downsampling has one big advantage. Filters operating on downsampled images have a bigger receptive field, that allows them to gather more context. This is especially important when trying to differentiate between classes occupying a small portion of the overall image, as, for example, rider and pedestrian in a road scene. It is just not enough that the network learns how people look, the context in which they appear is important as well. At the end, we have found that it is better to use dilated convolutions for the purpose of extending context information (Yu & Koltun (2015)).

**Early downsampling:**    One crucial intuition to achieving good performance and real-time operation is realizing that processing large input frames is very expensive. This might sound very obvious, however many popular architectures (Hong et al. (2015); Badrinarayanan et al. (2015b)) do not pay much attention towards optimization of early stages of network, which are often the most expensive.

ENet first two blocks heavily reduce the input size, and use only a small set of feature maps. The idea behind it, is that visual information is highly redundant in space, and thus can be compressed into a more efficient representation. Also, our intuition is that the initial network layers should not be used speciflly only for classification. Instead, they should rather act as good feature extractors and preprocess the input for later portions of the network. This insight worked well in our experiments. Increasing the number of feature maps from 16 to 32 did not improve the accuracy on Cityscapes dataset (Cordts et al. (2016)).

**Decoder size:**    In this work we would like to provide a different view on encoder-decoder architectures than the one presented in Badrinarayanan et al. (2015b). SegNet is a very symmetric architecture, as the encoder is an exact mirror of the encoder. Instead, our architecture consists of a large encoder, and a small decoder. This is motivated by the idea that the encoder should be able to work in a similar fashion to original classification architectures, i.e. to operate on smaller resolution data and provide for information processing and filtering. Instead, the role of the the decoder, is only to upsample the output of the encoder, fine-tuning the details.

**Nonlinear operations:**    He et al. (2016) report that it is beneficial to use ReLU and Batch Normalization layers before convolutions. We tried applying these ideas to ENet, but this had a detrimental effect on accuracy. Investigating its cause we replaced all ReLUs in the network with PReLUs (He et al. (2015a)), which use an additional parameter per feature map, with the goal of learning the negative slope of non-linearities. We expected that in layers where identity is a preferable transfer function, PReLU weights will have values around $1$, and conversely, values around $0$ if ReLU is preferable. Results of this experiment can be seen in Figure 2.

The first layers weights exhibit a large variance and are slightly biased towards positive values, while in the later portions of the encoder they settle to recurring pattern. All layers in the main branch behave nearly exactly like regular ReLUs, while the weights inside bottleneck modules are negative i.e. the function inverts and scales down negative values. We hypothesize that identity did not work out well in our architecture because of its limited depth. We hypothesize that the reason why such lossy functions are learned is that He et al. (2016) uses networks that are hundreds of layers deep, while our network uses fewer layers, and it needs to quickly filter out information. It is notable that the decoder weights become much more positive and learn functions closer to identity. This confirms our intuitions that the decoder is used only to fine-tune the upsampled output.

**Information-preserving dimensionality changes:**    As stated earlier, it is necessary to downsample the input early, but aggressive dimensionality reduction can also hinder the information flow. A very good approach to this problem has been presented in Szegedy et al. (2015b). However, pooling after a convolution, in case of increasing feature map depth, is computationally expensive. Therefore, we prefer to perform pooling operation in parallel with convolution of stride 2, and concatenate resulting feature maps. This technique allowed us to speed up inference time of the initial block 10 times.

Additionally, we have found one problem in the original ResNet architecture. When downsampling, the first $1 \times 1$ projection of the convolutional branch is performed with a stride of 2 in both dimensions, which effectively discards $75\%$ of the input. Increasing the filter size to $2 \times 2$ allows to take the full input into consideration, and thus improves the information flow and accuracy. Of course, it makes these layers $4\times$ more computationally expensive, however there are so few of these in ENet, that the overhead is not noticeable.

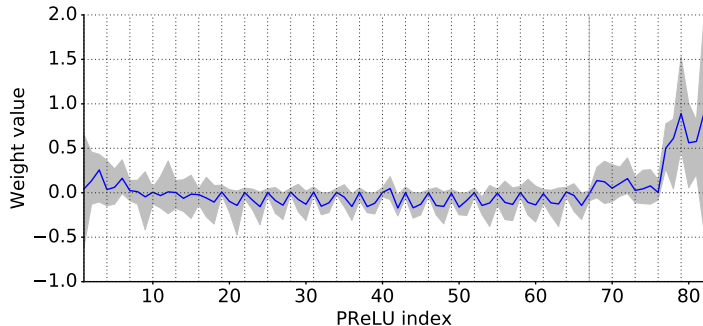

Figure 2: PReLU weight distribution vs network depth. Blue line is the weights mean, while an area between maximum and minimum weight is grayed out. Each vertical dotted line corresponds to a PReLU in the main branch and marks the boundary between each of bottleneck blocks. The gray vertical line at 67th module is placed at encoder-decoder border.

**Factorizing filters:** It has been shown that convolutional weights have a fair amount of redundancy, and each $n \times n$ convolution can be decomposed into two smaller asymmetric convolutions: one with a $n \times 1$ filter followed by a $1 \times n$ filter (Jin et al. (2014); Szegedy et al. (2015b)). We have used asymmetric convolutions with $n = 5$ in our network, so cost of these two operations is similar to a single $3 \times 3$ convolution. This allowed to increase the variety of functions learned by each block and increase the receptive field.

Sequence of operations used in the bottleneck module (projection, convolution, projection) can be seen as decomposing one large convolutional layer into a series of smaller and simpler low-rank approximation operations. Such factorization allows for large speedups and reduction in number of parameters, making them less redundant (Jin et al. (2014)).

**Dilated convolutions:** As argued above, it is very important for the network to have a wide receptive field, so it can perform classification by taking a bigger portion of the image (context) into account. We wanted to avoid overly downsampling the feature maps, and decided to use dilated convolutions (Yu & Koltun (2015)) to improve our model. We have used them inside several bottleneck modules, in particular the ones that operate on the smallest resolutions. These gave a significant accuracy boost, by raising IoU on Cityscapes by around 4 percentage points, with no additional cost. We obtained the best accuracy when we interleaved them with other bottleneck modules (both regular and asymmetric), instead of arranging them in sequence, as has been done in Yu & Koltun (2015).

**Regularization:** Most pixel-wise segmentation datasets are relatively small (on order of $10^3$ images), so such expressive models as neural networks quickly begin to overfit them. In initial experiments, we used L2 weight decay with little success. Then, inspired by Huang et al. (2016), we have tried stochastic depth, which increased accuracy. However it became apparent that dropping branches (i.e. setting their output to 0) is in fact a special case of applying Spatial Dropout (Tompson et al. (2015)), where either all of the channels, or none of them are ignored, instead of selecting a random subset. We placed Spatial Dropout at the end of convolutional branches, right before the addition, and it turned out to work much better than stochastic depth.

## 5 RESULTS

We benchmarked the performance of ENet on three different datasets to demonstrate real-time and accurate for practical applications. We tested on CamVid and Cityscapes datasets of road scenes, and SUN RGB-D dataset of indoor scenes. We set SegNet (Badrinarayanan et al. (2015b)) as a baseline since it is one of the fastest segmentation model available, that also requires less memory to operate than plain CNNs. All our models, training, testing and performance evaluation scripts were written using the Torch7 machine-learning library. To compare results, we use class average accuracy and intersection-over-union (IoU) metrics.

## 5.1 Performance Analysis

We report results on inference speed on widely used NVIDIA Titan X GPU as well as on NVIDIA TX1 embedded system module. ENet was designed to achieve more than 10 fps on the NVIDIA TX1 board with an input image size $640 \times 360$ (W,H), which is adequate for practical road scene parsing applications. For inference we merge batch normalization and dropout layers into the convolutional filters, to speed up all networks.

Table 2: Performance comparison. Image size is W×H

| Model | NVIDIA TX1 | | | | | | NVIDIA Titan X | | | | | |
|---|---|---|---|---|---|---|---|---|---|---|---|---|
| | 480×320 | | 640×360 | | 1280×720 | | 640×360 | | 1280×720 | | 1920×1080 | |
| | ms | fps | ms | fps | ms | fps | ms | fps | ms | fps | ms | fps |
| SegNet | 757 | 1.3 | 1251 | 0.8 | - | - | 69 | 14.6 | 289 | 3.5 | 637 | 1.6 |
| ENet | 47 | 21.1 | 69 | 14.6 | 262 | 3.8 | 7 | 135.4 | 21 | 46.8 | 46 | 21.6 |

**Inference time:**    Table 2 compares inference time for a single input frame of varying resolution. We also report the number of frames per second that can be processed. Dashes indicate that we could not obtain a measurement, due to lack of memory. ENet is significantly faster than competing architectures, providing high frame rates for real-time applications and allowing for practical use of very deep neural network models with encoder-decoder architecture.

Table 3: Hardware requirements. FLOPs are estimated for an input of $3 \times 640 \times 360$ (C,W,H).

| | GFLOPs | Parameters | Model size (fp16) |
|---|---|---|---|
| SegNet | 286.03 | 29.46M | 56.2 MB |
| ENet | 3.83 | 0.37M | 0.7 MB |

**Hardware requirements:**    Table 3 reports a comparison of number of operations and parameters used by different models. ENet efficiency is evident in the much low number of operations per frame and overall parameters. Please note that we report storage required to store the models in half precision floating point format. ENet has so few parameters that it can be saved into a file of just 0.7MB, which makes it possible to fit the whole network in an extremely fast on-chip memory in embedded processors. This alleviates the need for model compression (Han et al. (2015)), making it possible to use general purpose code for neural network computation. However, if one needs to operate under incredibly strict memory constraints, these techniques can still be applied to ENet as well.

**Software limitations:**    One of the most important techniques that has allowed us to reach these levels of performance is convolutional layer factorization. However, we have found one surprising drawback. Although applying this method allowed us to greatly reduce the number of floating point operations and parameters, it also increased the number of individual kernels calls, making each of them smaller.

We have found that some of these operations become so cheap, that the cost of GPU kernel launch starts to outweigh the cost of the actual computation. Also, because kernels do not have access to values that have been kept in registers by previous ones, they have to load all data from global memory at launch, and save it when their work is finished. This means that using a higher number of kernels, increases the number of memory operations, because feature maps have to be constantly saved and reloaded. This becomes especially apparent in case of non-linear operations. In ENet, PReLUs consume more than a quarter of inference time. Since they are only simple point-wise operations and very easy to parallelize, we hypothesize it is caused by the aforementioned data movement.

These are serious limitations, however they could be resolved by performing kernel fusion in existing software i.e. create kernels that apply non-linearities to results of convolutions directly, or perform a

number of smaller convolutions in one call. This improvement in GPU libraries, such as CuDNN, could increase the speed and efficiency of our network even further.

## 5.2 BENCHMARKS

During training we have used the Adam optimization algorithm (Kingma & Ba (2014)). It allowed ENet to converge very quickly and on every dataset we haver used training took only 3-4 hours on Titan X. Training of ENet was performed in two stages: first we train only the encoder to categorize downsampled regions of the input image, then we appended the decoder and train the network to perform upsampling and pixel-wise categorization. in this work, a learning rate of $5e-4$ and L2 weight decay of $2e-4$, along with batch size of $10$ consistently provided the best results. For categorization, we have used a custom class weighing scheme defined as $w_{\text{class}} = \frac{1}{\ln(c+p_{\text{class}})}$. In contrast to the inverse class probability weighing, the weights are bounded as the probability approaches $0$. $c$ is an additional hyper-parameter, which we set to $1.02$ (i.e. we restrict the class weights to be in the interval of $[1, 50]$).

Table 4: Cityscapes test set results

| Model | Class IoU | Class iIoU | Category IoU | Category iIoU |
|---|---|---|---|---|
| SegNet | 56.1 | 34.2 | 79.8 | **66.4** |
| ENet | **58.3** | **34.4** | **80.4** | 64.0 |

**Cityscapes:** This dataset consists of 5000 fine-annotated images, out of which 2975 are available for training, 500 for validation, and the remaining 1525 have been selected as test set (Cordts et al. (2016)). Cityscapes was the most important benchmark for us, because of its outstanding quality and highly varying road scenarios, often featuring many pedestrians and cyclists. We trained on 19 classes that have been selected in the official evaluation scripts (Cordts et al. (2016)). It makes use of an additional metric called instance-level intersection over union metric (iIoU), which is IoU weighed by the average object size. As reported in Table 4, ENet outperforms SegNet in class IoU and iIoU, as well as in category IoU. ENet is currently the fastest model in the Cityscapes benchmark.

Table 5: Results on CamVid test set of (1) SegNet-Basic, (2) SegNet, and (3) ENet

| Model | Building | Tree | Sky | Car | Sign | Road | Pedestrian | Fence | Pole | Sidewalk | Bicyclist | Class avg. | Class IoU |
|---|---|---|---|---|---|---|---|---|---|---|---|---|---|
| 1 | 75.0 | 84.6 | 91.2 | **82.7** | 36.9 | 93.3 | 55.0 | 47.5 | **44.8** | 74.1 | 16.0 | 62.9 | n/a |
| 2 | **88.8** | **87.3** | 92.4 | 82.1 | 20.5 | **97.2** | 57.1 | 49.3 | 27.5 | 84.4 | 30.7 | 65.2 | **55.6** |
| 3 | 74.7 | 77.8 | **95.1** | 82.4 | **51.0** | 95.1 | **67.2** | **51.7** | 35.4 | **86.7** | **34.1** | **68.3** | 51.3 |

**CamVid:** Another automotive dataset, on which we have tested ENet, was CamVid. It contains 367 training and 233 testing images (Brostow et al. (2008)). There are eleven different classes such as building, tree, sky, car, road, etc. while the twelfth class contains unlabeled data, which we ignore while training. The original frame resolution for this dataset is 960×720 (W,H) but we downsampled the images to 480×360 before training. In Table 5 we compare the performance of ENet with existing state-of-the-art algorithms. ENet outperforms other models in six classes, which are difficult to learn because they correspond to smaller objects.

Table 6: SUN RGB-D test set results

| Model | Global avg. | Class avg. | Mean IoU |
|---|---|---|---|
| SegNet | **70.3** | **35.6** | **26.3** |
| ENet | 59.5 | 32.6 | 19.7 |

**SUN RGB-D:**  The SUN dataset consists of 5285 training images and 5050 testing images with 37 indoor object classes. We did not make any use of depth information in this work and trained the network only on RGB data. In Table 6 we compare the performance of ENet with SegNet (Badrinarayanan et al. (2015b)), which is the only neural network model that reports accuracy on this dataset. Our results, though inferior in global average accuracy and IoU, are comparable in class average accuracy. Since global average accuracy and IoU are metrics that favor correct classification of classes occupying large image patches, researchers generally emphasize the importance of other metrics in case of semantic segmentation. One notable example is introduction of iIoU metric (Cordts et al. (2016)). Comparable result in class average accuracy indicates, that our network is capable of differentiating smaller objects nearly as well as SegNet. Moreover, the difference in accuracy should not overshadow the huge performance gap between these two networks. ENet can process the images in real-time, and is nearly $20\times$ faster than SegNet on embedded platforms.

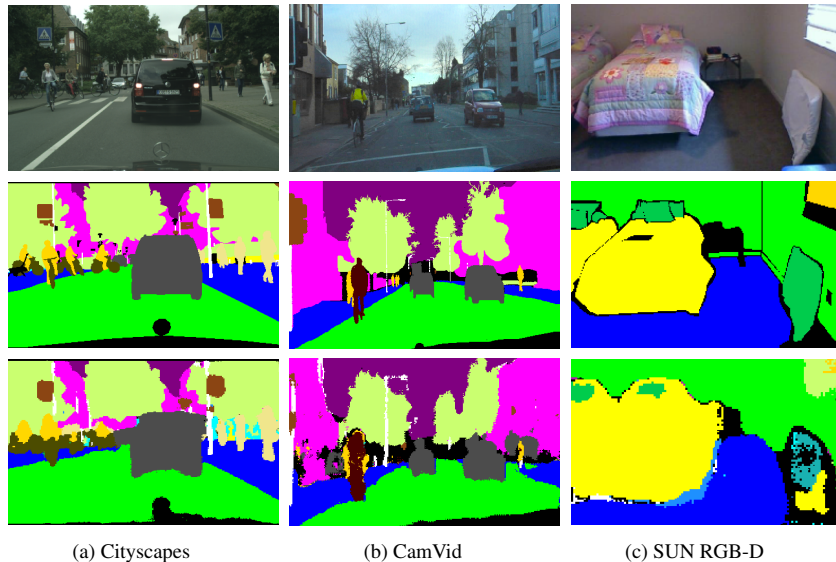

(a) Cityscapes                    (b) CamVid                    (c) SUN RGB-D

Figure 3: ENet predictions on popular benchmarks (rows top to down represent input image, ground truth, and ENet output respectively).

# 6  CONCLUSION

We have proposed a novel neural network architecture designed from the ground up specifically for semantic segmentation. Our main aim is to make efficient use of scarce resources available on embedded platforms, compared to fully fledged deep learning workstations. Our work provides large gains in this task, while matching and at times exceeding existing baseline models, that have an order of magnitude larger computational and memory requirements. The application of ENet on the NVIDIA TX1 hardware exemplifies real-time portable embedded solutions.

Even though the main goal was to run the network on mobile devices, we have found that it is also very efficient on high end GPUs like NVIDIA Titan X. This may prove useful in data-center applications, where there is a need of processing large numbers of high resolution images. ENet allows to perform large-scale computations in a much faster and more efficient manner, which might lead to significant savings.

ACKNOWLEDGMENT

This work is partly supported by the Office of Naval Research (ONR) grants N00014-12-1-0167, N00014-15-1-2791 and MURI N00014-10-1-0278. We gratefully acknowledge the support of NVIDIA Corporation with the donation of the TX1, Titan X, K40 GPUs used for this research.

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
