# Peer review of "ENet: A Deep Neural Network Architecture for Real-Time Semantic Segmentation"

_ICLR 2017 — rejected_

[Public Comment · (anonymous) · 14 Nov 2016]
**Comparison on VOC?**

Interesting work and especially relevant going forward with the plethora of mobile devices. It would be especially interesting to see the time comparisons
on a mobile device using any of the currently available mobile frameworks. For example iOS has CNN support using Metal. Here is sample code [bit.ly/2fQQcrX]
that specifically runs VGG on an iPhone (not in fully convolutional format) but which you could probably modify easily to match your architecture.
Also, have you tried training the same architecture on the Pascal VOC segmentation challenge? SegNet has a mean IU of 59.9 according to the VOC benchmark [bit.ly/2g9Mpa3]. What does ENet achieve?

[Public Comment · (anonymous) · 03 Dec 2016]
**good project yet still of low technical quality ...**

I have tried the open source implementation for different tasks these several months. It's fast, reasonably accurate and useful for prototyping.

However I see no technical quality improvements against its NIPS submission version.

I am really wondering which design choice is the most dominant one. 

Wondering what I should do if I want to design a network as efficient as this one.

[Official Review · AnonReviewer3 · rating 3 · confidence 4 · 16 Dec 2016]
**This may be of interest to practitioners - but I do not think that there is a clear academic value to this paper.**

The paper introduces a lightweight network for semantic segmentation that combines several acceleration ideas.
As indicated in my preliminary question, the authors do not make the case about why any of the techniques they propose is beyond what we know already: factorizing filters into alternating 1-D convolutions, using low-rank kernels, or any of the newer inception network architectures.

I have had a hard time figuring out what is the take-home message of this paper. All of these ideas are known, and have proven their worth for detection. If a paper is going to be accepted for applying them to semantic segmentation, then in the next conference another paper should be accepted for applying them to normal estimation, another to saliency estimation and so on. 

As the authors mention in their preliminary review:
"I agree that most improvements from classification architectures are straightforward to apply to object segmentation, and that's exactly what we've done - our network is based on current state of the art models. Instead of repeating most of the discussion on factorizing filters, etc., that has been discussed in a lot of papers already, we have decided that it's much more valuable to describe in depth the choices that are related to segmentation only - these are the most important contributions of our paper."

I do not see however any in-depth discussion of certain choices - e.g. an analysis of how certain choices influence performance or speed. Instead all one gets are some statements "these gave a significant accuracy boost" "this helped a lot", "that did not help", "this turned out to work much better than that" . This is not informative - and is more like an informal chat rather than an in-depth discussion. 

If novelty is not that important, and it is only performance or speed that matter, I am still not convinced.
The authors only compare to [1,2] (SegNet) in terms of both accuracy and speed. I cannot see the reason why they do so, and they do not really justify it. According to the authors' evaluation, [1] requires ~1 sec. per frame,  while Deeplab v2, without the DenseCRF, runs at 5-8fps. 
(

[Official Review · AnonReviewer2 · rating 5 · confidence 4 · 17 Dec 2016]
**fast system, but needs more thorough account**

This paper describes a fast image semantic segmentation network.  Many different techniques are combined to create a system much faster than the baseline SegNet approach, with accuracy comparable or somewhat worse in most of three datasets evaluated.

The choices and techniques used to achieve these speed optimizations are enumerated and described along with intuitions behind them.  However, this section lacks measurements and experimental results showing the effects of these choices.  To me, that would have been a key component to the paper.  As it stands now, we only get to see final evaluation numbers, which appear to describe a speed/accuracy tradeoff with little insight into the pieces sum to get there.

In addition, I feel there could be a more thorough comparison with different existing systems.  Only SegNet is shown in comparison tables, even though many current systems are outlined in the related work.  Additional datasets such as Pascal or COCO may be interesting here as well, perhaps with a larger version of the ENet model.

The system looks to be fast, with decent accuracy on the majority of benchmarks described.  However, as a practical implementation paper, I feel it needs to more thoroughly demonstrate the effects of each component, as well as possibly some of the sizing/tuning, in order to provide a more robust picture.

[Official Review · AnonReviewer4 · rating 4 · confidence 4 · 31 Dec 2016]
**Fast and compressed semantic segmentation. Lack of novelty. Experiments are not convincing enough.**

This paper aims at designing a real-time semantic segmentation network. The proposed approach has an encoder-decoder architecture with many pre-existing techniques to improvement the performance and speed. 

My concern is that the most of design choices are pretty ad-hoc and there is a lack of ablation study to validate each choice. 

Moreover, most of the components are not new to the community (indexed pooling, dilated convolution, PReLu, steerable convolution, spatial dropout). The so-called 'early down-sampling' or 'decoder size' are also just very straightforward trade-off between speed and performance through reducing the size/depth of the layers. 

The performance and inference comparison is only conducted against a rather weak baseline, SegNet, which also makes the paper less convincing. On the public benchmark the proposed model does not achieve comparable results against state-of-the-art. As some other reviewer raised, there are some stronger model that has similar efficiency compared with SegNet.

The speed-up improvement is good yet reasonable given all the components used. However, we also did see a big sacrifice in performance on some benchmarks, which makes all these tricks less promising. 

The only fact I found impressive is that the model size is 0.7MB, which is of good practical use and helpful to dump on mobile devices. However, there is NO analysis over how is the trade-off between the model size and the performance, and what design would result how much reduction in model size. I did not find the memory consumption report for the inference stage, which are perhaps even more crucial for embedded systems. 

Perhaps this paper does have a practical value for practical segmentation network design on embedding systems. But I do not believe the paper brings insightful ideas that are worthy to be discussed in ICLR, either from the perspective of model compression or semantic segmentation.

[Official Review · AnonReviewer1 · rating 4 · confidence 4 · 06 Jan 2017]
**Interesting work, lack of depth**

Paper summary: this work presents ENet, a new convnet architecture for semantic labeling which obtains comparable performance to the previously existing SegNet while being ~10x faster and using ~10x less memory. 


Review summary: Albeit the results seem interesting, the paper lacks detailed experimental results, and is of limited interest for the ICLR audience.


Pros:
* 10x faster
* 10x smaller
* Design rationale described in detail


Cons:
* The quality of the reference baseline is low. For instance, cityscapes results are 58.3 IoU while state of the art is ~80 IoU. Thus the results are of limited interest.
* The results that support the design rationale are not provided. It is important to provide the experimental evidence to support each claim.


Quality: the work is interesting but feels incomplete. If your model is 10x faster and smaller, why not try build a model 10x longer to obtain improved results ? The paper focuses only on  nimbleness at the cost of quality (using a weak baseline). This limits the interest for the ICLR audience.


Clarity: the overall text is somewhat clear, but the model description (section 3) could be more clear. 


Originality: the work is a compendium of “practitioners wisdom” applied to a specific task. It has thus limited originality.


Significance: I find the work that establishes a new “best practices all in one” quite interesting, but however these must shine in all aspects. Being fast at the cost of quality, will limit the impact of this work.


Minor comments:
* Overall the text is proper english but the sentences constructions is often unsound, specific examples below. 
* To improve the chances of acceptance, I invite the authors to also explore bigger models and show that the same “collected wisdom” can be used both to reach high speed and high quality (with the proper trade-off curve being shown). Aiming for only one end of the quality versus speed curve limits too much the paper.
* Section 1: “mobile or battery powered … require rates > 10 fps“. 10 fps with which energy budget ? Should not this be  > 10 fps && < X Watt.
* “Rules and ideas” -> rules seem too strong of a word, “guidelines” ?
* “Is of utmost importance” -> “is of importance” (important is already important)
* “Presents a trainable network … therefore we compare to … the large majority of inference the same way”; the sentence makes no sense to me, I do not see the logical link between before and after “therefore”
* Scen-parsing -> scene-parsing
* It is arguable if encoder and decoder can be called “separate”
* “Unlike in Noh” why is that relevant ? Make explicit or remove
* “Real-time” is vague, you mean X fps @ Y W ?
* Other existing architectures -> Other architectures
* Section 3, does not the BN layer include a bias term ? Can you get good results without any bias term ?
* Table 1: why is the initial layer a downsampling one, since the results has half the size of the input ?
* Section 4, non linear operations. What do you mean by “settle to recurring pattern” ?
* Section 4, dimensionality changes. “Computationally expensive”, relative to what ?
* Section 4, dimensionality changes. “This technique ... speeds-up ten times”, but does not provide the same results. Without an experimental validation changing an apple for an orange does not make the orange better than the apple.
* Section 4, dimensionality changes. “Found one problem”, problem would imply something conceptually wrong. This is more an “issue” or an “miss-match” when using ResNet for semantic labelling.
* Section 4, factorizing filters. I am unsure of why you call nx1 filter asymmetric. A filter could be 1xn yet be symmetric (e.g. -2 -1 0 1 2). Why not simply call them rectangular filters ?
* Section 4, factorizing filters. Why would this change increase the variety ? I would have expected the opposite.
* Section 4, regularization. Define “much better”.
* Section 5.1; “640x360 is adequate for practical applications”; for _some_ applications.
* Section 5.2, “very quickly” is vague and depends on the reader expectations, please be quantitative.
* Section 5.2, Haver -> have
* Section 5.2, in this work -> In this work
* Section 5.2, unclear what you use the class weighting for. Is this for class balancing ?
* Section 5.2, Cityscapes was -> Cityscapes is
* Section 5.2, weighted by the average -> is each instance weighted relative the average object size.
* Section 5.2, fastest model in the Cityscapes -> fastest model in the public Cityscapes

[Final Decision · Program Chairs · 06 Feb 2017]
**ICLR committee final decision**

Three knowledgable reviewers recommend rejection and there was no rebuttal. The AC agrees with the reviewers.